

# Feedbacks between earlywood anatomy and non-structural carbohydrates affect spring phenology and wood production in ring-porous oaks

Gonzalo Pérez-de-Lis[1], Ignacio García-González[1], Vicente Rozas[2], José Miguel Olano[2]

[1] Departamento de Botánica, Escola Politécnica Superior, Universidade de Santiago de Compostela, Lugo, 27002, Spain
[2] Área de Botánica, EUI Agrarias, Universidad de Valladolid, Soria, 42004, Spain

*Correspondence to*: Gonzalo Pérez-de-Lis (gonzalo.perezdelis@usc.es)

**Abstract.** Non-structural carbohydrates (NSC) play a central role in the construction and maintenance of the vascular system, but feedbacks between the NSC status of trees and wood formation are not fully understood. We aimed to evaluate multiple dependencies among wood anatomy, winter NSC, and phenology for coexisting temperate (*Quercus robur*) and sub-Mediterranean (*Q. pyrenaica*) oaks along a water-availability gradient in NW Iberian Peninsula. Sapwood NSC was quantified at three sites in December 2012 (n = 240). Leaf phenology and wood anatomy were surveyed in 2013. Structural equation modelling was used to analyze the interplay among hydraulic diameter ($D_h$), winter NSC, date of budburst, and earlywood vessel production (EVP), while the effect of $D_h$ and EVP on latewood width was assessed by using a mixed-effects model. NSC and wood production increased under drier conditions in both species. *Q. robur* showed narrower $D_h$ and lower soluble sugar (SS) concentration (3.88–5.08 % dry matter) than *Q. pyrenaica* (4.06–5.57 % dry matter), but *Q. robur* exhibited larger EVP and wider latewood (1,403 µm) than *Q. pyrenaica* (667 µm). Trees of both species with large $D_h$ showed higher SS concentration in winter and earlier flushing. *Q. pyrenaica* exhibited a carbon saving strategy, as evidences the fact that EVP was in tune with SS content in winter. Latewood production was controlled by $D_h$ and EVP, rather than by foliage density and growing season duration. Our results suggest that high SS content in oaks with high conductive area favours an earlier spring phenology, as well as earlywood growth. *Q. pyrenaica* exhibited a tighter control of carbohydrate allocation to xylem formation than *Q. robur*, which is probably related to the acquisition of physiological resistance to stress in the sub-Mediterranean area.

## 1 Introduction

Non-structural carbohydrates (NSC) have multiple key functions in trees, such as fuel maintenance respiration, osmoregulation, cryoprotection, or growth control (Morin et al., 2007; Sala et al., 2012; Wang and Ruan, 2013; Dietze et al., 2014). The asynchrony between carbon assimilation and consumption is solved by accumulating non-structural carbohydrate (NSC) reserves (Chapin et al., 1990; Sala et al., 2012), which are mostly stored in stem, branches and coarse roots as soluble sugars (SS) and starch (Barbaroux et al., 2003). A large part of the NSC budget of the tree is invested in construction and maintenance of the vascular system, as well as in fine roots turnover, and crown development (Wang and Ruan, 2013). The hydraulic network in ring-porous oaks is highly vulnerable to dysfunction due to cavitation of their large vessels, which operate at a narrow safety margin (Delzon and Cochard, 2014; Urli et al., 2015). The refilling of these embolized vessels needs restoration of osmotic gradients through releasing SS into the conduits (Salleo et al., 2009; Brodersen and McElrone, 2013). Alternatively, the hydraulic function can be recovered through the formation of new conduits the following spring (Brodribb et al., 2010). In ring-porous species, earlywood vessels are generally functional during only one year (Urli et al., 2015), and cambial resumption precedes leaf formation (Pérez-de-Lis et al., 2016). Large NSC reserves are therefore needed in order to provide energy and materials for leaf expansion and cambial activity at the onset of the growing season (Barbaroux et al., 2003; El Zein et al., 2011).





Tall trees are thought to form wider vessels at the tree base in order to compensate height-related hydraulic resistance in the stem (Petit et al., 2008). Probably, larger vessels boost carbon gain in dominant individuals since stomatal conductance increases with the hydraulic capacity (Fichot et al., 2009). In fact, fast-growing trees commonly exhibit larger NSC levels and a faster NSC turnover than their slow-growing counterparts (Sundberg et al., 1993; Sala and Hoch, 2009; Carbone et al.,

2013). Concurrently, more carbohydrates may be allotted to hydraulic purposes in trees with wider but more vulnerable vessels (Brodersen and McElrone, 2013). However, little is known about how feedbacks between wood anatomy and short-distance NSC mobilization affect growth.

Entangled relationships between tree vigour and growth are influenced by the timing of tree phenology, given that dominant trees show larger growing periods and more intense cambial activity (Rathgeber et al., 2011). Although the break

of dormancy is mostly controlled by temperature and photoperiod (Basler and Körner, 2014), some studies suggest that high NSC concentrations in developing buds speed up leaf-out dates (Maurel et al., 2004). In winter, phloem of deciduous trees was suggested to be non-functional, whereby sapwood might be involved in carbon translocation through the plant body (Lacointe et al., 2004). The influx of sucrose from xylem conduits in branches into the buds was reported to be tightly correlated to bud swelling rates (Bonhomme et al., 2010), whilst high sucrose concentrations in the stem of mutant poplars

have been associated to advanced budburst (Park et al., 2009). In ring-porous oaks, winter temperature has been reported to affect earlywood formation, which has been attributed to direct effects of temperature on both the timing of phenology and carbon demand for maintenance respiration (Gea-Izquierdo et al., 2012). However, the influence of sapwood NSC levels in dormancy release is frequently precluded, and the possible effect of their interplay in tree growth is poorly understood.

There is a tight association between the timing of primary and secondary growth at the whole-tree level (Pérez-de-Lis

et al., 2016). In turn, ecological requirements modulating phenology are linked to functional species-specific strategies (Basler and Körner, 2014). This is the case of the ring-porous oaks *Quercus robur* L. and *Q. pyrenaica* Willd., which coexist in NW Iberian Peninsula. The latter is a sub-Mediterranean species that exhibits late flushing (Pérez-de-Lis et al., 2016), along with several morphological and physiological adaptations to cope with summer drought and winter frost. Such differences could impact carbon metabolism and allocation to growth (Valladares et al., 2000; Piper, 2011; Guillemot et al.,

2015), as well as the rate of developmental processes (Deslauriers et al., 2009), affecting the adaptive capacity to track rapid climate change (Jump and Peñuelas, 2005). Water shortage is deemed to influence carbon metabolism in a complex manner by constraining the activity of both source and sink organs (Sala et al., 2012). Whereas some studies reported that NSC are accumulated under drought (Sala and Hoch, 2009; Lempereur et al., 2015), other authors found a drought-induced reduction in starch concentration, coupled with changes in the SS composition (Rosas et al., 2013; Deslauriers et al., 2014). Therefore,

we need to understand how phenology and growth are coordinated with NSC and wood anatomy in order to better predict plant responses to climate in the context of global warming.

In this research, data from sapwood NSC concentration in winter 2012, xylem anatomical traits, and phenology in spring 2013 are used to disentangle the influence of their mutual interactions on the radial growth of ring-porous oaks growing along a water-availability gradient in NW Iberian Peninsula. We focused on the possible differences in xylem

anatomy and NSC between *Q. robur* and *Q. pyrenaica* along the gradient. In this regard, we hypothesized that the more drought-tolerant *Q. pyrenaica* will have a larger NSC pool and more reduced growth than *Q. robur*. Wood production and NSC concentration are expected to vary along the gradient, with reduced growth but increasing NSC storage under drier conditions. At the species level, we aim to test the following hypotheses: (i) tree size influences earlywood vessel diameter, which in turn affects NSC content; (ii) higher SS content in winter predisposes trees to advance growth resumption in spring,

thereby fostering earlywood production; and (iii) earlywood anatomical features are important predictors of latewood growth in oaks.



## 2 Materials and methods

### 2.1 Study sites

The study area is located in NW Iberian Peninsula, at the transition between the Atlantic and Mediterranean biogeographical regions (Fig. 1a). Three mixed stands of *Q. robur* (hereafter *Qrob*) and *Q. pyrenaica* (hereafter *Qpyr*) were selected along a

north-to-south gradient of decreasing water supply (Fig. 1B). Annual rainfall declines from 1,461 mm at the northernmost site Bermui (hyperhumid), to 996 mm at Labio (humid), and to 832 mm at the southernmost site Moreiras (subhumid). Mean annual temperature is lower at the hyperhumid (11.3 ºC) and humid (11.6 ºC) sites than at the subhumid site (14.4 ºC). Sampled stands are dominated by *Qrob* at the hyperhumid and humid sites, whereas *Qpyr* is more frequent at the subhumid location. Forests at the hyperhumid and humid sites include temperate trees and understory shrubs, such as *Betula alba* L.,

*Castanea sativa* Mill.*, Pyrus cordata* Desy.*, Ilex aquifolium* L.*, Daboecia cantabrica* (Huds.) K. Koch*,* and *Vaccinium myrtillus* L. By contrast, thermophilic Mediterranean flora, such as *Q. suber* L., *Laurus nobilis* L., *Arbutus unedo* L., *Osyris alba* L., and *Daphne gnidium* L., dominates the subhumid location. Stand tree densities are 1,178, 1,082, and 530 ha$^{-1}$ at the hyperhumid, humid and subhumid sites, respectively. Drought episodes can appear at the study region in summer, particularly at the subhumid site, which is the driest location within the gradient (Fig. 1b). Incident rainfall in 2012 at the

hyperhumid, humid, and subhumid locations was respectively 8, 14, and 33 % lower than the 1981–2010 average, whereas it was 35, 36, and 3 % higher than the average in 2013 (Table 1). However, in summer 2013, there were only 34 (45 mm), 11 (20 mm), and 8 (35 mm) days of precipitation at the hyperhumid, humid and subhumid locations, respectively. Furthermore, mean maximum temperature in the same period was 22.0 ºC at the hyperhumid and humid sites, but 25.3 ºC at the subhumid location.

### 2.2 Phenology and NSC concentration

At each study site, 40 trees per species were selected for sampling (overall n = 240). Stem diameter was measured for each tree in October 2012. Leaf phenology was weekly monitored during 2013 using binoculars (10×) at *ca.* 10 m distance from each tree. Budburst was identified as the day of year (DOY) in which the apical buds on the uppermost part of the canopy were green and expanding, but no leaves were distinguishable yet. In late autumn, leaf shedding was identified as the date in

which at least 50% of the leaves were shed from the crown. In addition, foliage density was visually estimated in July 2013 by counting the proportion of gaps in the crown, being expressed as a percentage of the theoretical maximum foliage density.

In order to analyze feedbacks between NSC and xylem anatomy, we quantified the content of NSC in sapwood by sampling one 5-mm diameter wood core per tree with an increment borer at breast height. Cores were taken in mid-December 2012, soon after the completion of leaf abscission, which occurred between mid and late November for both

species. After extraction, cores were immediately placed into a cool box, and subsequently stored at –20 ºC to prevent carbohydrate degradation. Before NSCs extraction, bark and traces of heartwood were removed, and the cores were oven-dried at 60 ºC for 72 hours. Sapwood was then finely grounded with a mixer mill (Retsch MM 400, Düsseldorf, Germany). We quantified NSC concentration for the whole sapwood by using the anthrone method (Olano et al., 2006). SS were extracted from 20 mg of dry mass in 1 ml of ethanol (80%) at 80 ºC for 30 min. The extract was centrifuged 10 min at 4,000

rpm, and the supernatant was collected for the spectrophotometrical determination of SS concentrations, for which we used the anthrone reagent. Starch contained in the residue was hydrolized with 1 ml of perchloric acid (35%) for 1 hour, and determination was conducted by using the anthrone reagent, as previously described for SS. Total NSC, SS, and starch concentrations were expressed as percentage of dry matter.



### 2.3 Wood anatomical measurements

One additional core per tree was collected in October 2013 to perform wood anatomical measurements. Cores were air-dried and mounted on wooden supports to be cut using a microtome (WSL Core Microtome, Zurich, CH) and polished. Cross-sectional surfaces were photographed with a digital camera (Canon EOS 600D, Tokyo, Japan), attached to a transmitted light microscope (Olympus BX40, Tokyo, Japan). Image analysis was applied on the rings formed in 2012 and 2013 using ImageJ 1.48v (Schneider et al., 2012), in order to quantify the lumen area of earlywood vessels, latewood width, and the number of earlywood vessels, which is a *proxy* of earlywood vessel production (EVP). For each vessel, we estimated the diameter of the equivalent circle, obtaining the hydraulic diameter ($D_h$) at the tree level according to the following Eq.:

$$D_h = \frac{\sum_{n=1}^{N} d_n^5}{\sum_{n=1}^{N} d_n^4}, \tag{1}$$

where $d_n$ is the diameter of the $n$ vessel (Sperry et al., 1994). According to the Hagen-Poiseuille equation, $D_h$ is proportional to the hydraulic capacity.

### 2.4 Comparisons along the gradient

Variation among sites and between species for NSC, dates of budburst and leaf shedding, wood anatomical traits, and foliage density were evaluated by applying generalized linear models (GLM) for gamma-distributed variables. Multiple pairwise comparisons were also assessed to test differences among site factor levels. This analysis was performed by using the packages 'lme4' and 'multcomp' for R 3.1.1 (R Core Team, 2014).

### 2.5 Connections among earlywood anatomy, sapwood NSC content and spring phenology

We performed structural equation models (SEM) to disentangle, at the species level, the role of winter NSC as possible regulators of budburst and EVP in 2013. Thereby, data from all the sites were pooled, and a unique model was fit for each species. SEM approach provides an adequate representation for interacting systems, in which simultaneous influences and responses including direct and indirect effects are explored (Grace, 2006). The structure of a hypothetical SEM, and its calculation, requires incorporating available *a priori* knowledge. According to the lines of evidence showed in the introduction, we hypothesized that fast-growing trees (larger stem diameter) show higher SS and starch concentrations due to their larger $D_h$ (Supplement, Fig. S1). In turn, high SS and starch concentrations in winter are expected to speed up tree phenology (date of budburst) and boost EVP during the subsequent year.

Standardized coefficients were estimated by the maximum likelihood method, and model evaluation was performed using a $\chi^2$ test. A $P$-value below 0.05 indicates that discrepancy between observed and expected covariance matrices is acceptable. The adjusted goodness of model fit index (AGFI), and the root mean square error of approximation (RMSEA) were complementarily performed in order to consider the effect of sample size on the model fit evaluation. Values of AGFI and RMSEA respectively above 0.90 and below 0.05 indicate an acceptable fit of the model in relation to the degrees of freedom. A $\chi^2$ test for multi-group invariance was applied to evaluate differences between the models fitted for each species. SEM analyses were performed with AMOS 18.0 software (AMOS Development Corp., Mount Pleasant, South Carolina, USA).

### 2.6 Predictors of latewood formation

We performed generalized linear mixed-effects models (GLMM) to identify which factors affected latewood production in 2013 at the species level. The effect of site was included as a random component, while winter NSCs, earlywood anatomy ($D_h$ and EVP in 2013), growing season length, and foliage density were the explanatory variables of the model. Collinearity was surveyed by calculating the generalized variance-inflation factors for each species. GLMM models were fitted by a log-link function with a gamma distribution, being ranked according to the second-order Akaike's Information Criterion (Bolker



et al., 2009). We averaged the 95% confidence set of models according to the Akaike weights, and the relative importance of a given variable was calculated as the sum of the Akaike weights across all the models in which it was contained (Burnham and Anderson, 2002). We used the packages 'lme4' and 'MuMIn' for R 3.1.1 (R Core Team, 2014) to assess GLMMs.

## 3 Results

### 3.1 Variation in NSC, wood anatom and leaf phenology along the gradient

Mean SS concentrations at the sites ranged from 3.88 to 5.08 % dry matter in *Qrob*, and from 4.06 to 5.57 % dry matter in *Qpyr* (Fig. 2), being similar to those of starch, which ranged from 4.28 to 5.11 % dry matter in *Qrob*, and from 3.47 to 5.11 % in *Qpyr* (Fig. 2). *Qpyr* exhibited therefore greater SS content than *Qrob* ($F = 18.27$, $P < 0.001$), while both starch and NSC concentration did not differ between species, although were marginally significant for NSC ($F_{starch} = 2.14$, $F_{NSC} = 0.62$, $P > 0.050$). Such pattern resulted in a higher SS-to-starch ratio for *Qpyr* than for *Qrob* ($F = 18.07$, $P < 0.001$), especially at the hyperhumid and humid locations (Fig. 2), whereas there was no variation along the gradient ($F = 0.21$, $P = 0.814$). In contrast, NSC substantially varied among locations ($F = 22.34$, $P < 0.001$), with decreasing concentrations from the subhumid to the hyperhumid sites (Fig. 2). SS content followed a similar geographical pattern, in both *Qrob* ($F = 17.72$, $P < 0.001$) and *Qpyr* ($F = 21.89$, $P < 0.001$). The subhumid site exhibited a higher starch content than the hyperhumid location for *Qpyr* ($F = 8.59$, $P < 0.001$), whereas no clear geographical pattern was found for *Qrob* ($F = 2.52$, $P = 0.085$).

Overall, *Qpyr* exhibited a higher $D_h$ than *Qrob* ($F_{2012} = 7.76$, $F_{2013} = 8.31$, $P < 0.010$). Yet, *Qrob* had larger EVP ($F = 30.28$, $P < 0.001$) and wider latewood ($F = 51.15$, $P < 0.001$) than *Qpyr* (Fig. 3a, b). In *Qpyr*, $D_h$ was substantially lower at the hyperhumid site in 2012 ($F = 7.67$, $P < 0.001$) and 2013 ($F = 3.72$, $P = 0.027$), whereas much less variation was found among sites for *Qrob* ($F_{2012} = 2.89$, $F_{2013} = 0.18$, $P > 0.050$). EVP and latewood width differed along the gradient ($F_{EVP} = 8.45$, $F_{LW} = 12.50$, $P < 0.001$), with the lowest values at the hyperhumid site (Fig. 3c, d). Conversely, the highest EVP and latewood width values occurred at the subhumid site for *Qrob*, and at the humid site for *Qpyr* (Fig. 3c, d).

Stem diameter was positively correlated with tree height in *Qrob* ($r = 0.60$, $P < 0.001$) and *Qpyr* ($r = 0.58$, $P < 0.001$). At the hyperhumid site, stem diameter was larger for *Qrob* than for *Qpyr* ($Z = 3.29$, $P = 0.012$), whereas the studied species showed similar values at the humid and subhumid locations (Fig. 4a). Trees at the subhumid site had a larger stem diameter than at the humid (*Qrob* $Z = 5.08$, *Qpyr* $Z = 7.58$, $P < 0.001$) and hyperhumid locations (*Qrob* $Z = 2.67$, $P = 0.021$; *Qpyr* $Z = 7.13$, $P < 0.001$). *Qpyr* exhibited a later budburst than *Qrob* ($F = 527.83$, $P < 0.001$), occurring from mid to late April at the subhumid site, and from late April to late May at the hyperhumid and humid locations (Fig. 4b). Budburst occurred synchronously at the hyperhumid and humid sites (*Qrob* $t = 1.92$, *Qpyr* $t = –0.54$, $P > 0.05$), but comparatively earlier (*Qrob* $F = 128.45$, *Qpyr* $F = 79.49$, $P < 0.001$) at the subhumid location (Fig. 4b). Trees showing an earlier budburst had a delayed senescence (*Qrob* $r = –0.36$, *Qpyr* $r = –0.68$, $P < 0.001$). Leaf shedding was first recorded at the hyperhumid and humid sites for *Qpyr* (DOY 312 on average), whereas some green leaves could be perceived until late December at the subhumid site (*Qrob* DOY 357, *Qpyr* DOY 354; Fig. 4c). The period from budburst to leaf shedding in 2013 was on average 42 days longer for *Qrob* than for *Qpyr* ($F = 450.90$, $P < 0.001$). Foliage density was similar along the gradient for *Qrob*, but significantly lower at ALT for *Qpyr* (Fig. 4d). It is also relevant to consider that numerous *Qpyr* trees at this latter location had their leaves infected with powdery mildew in spring 2013.

### 3.2 Species-specific models on functional relationships affecting wood production

SEM models showed a good fit for both species ($df = 1$; *Qrob*, $\chi^2 = 0.202$, $P = 0.653$; *Qpyr*, $\chi^2 = 0.118$, $P = 0.732$), with AGFI > 0.90 and RMSEA < 0.1 (Fig. 5a, b). Large trees exhibited a higher $D_h$ in 2012, having a positive indirect effect on SS levels, irrespective of the species. SS concentration in December showed positive covariation with starch content in both species (Fig. 5). High SS contents were associated to advanced budburst, whereas starch concentration and date of budburst were unrelated. In *Qpyr*, SS concentrations had a positive direct effect in EVP. By contrast, this relationship was not direct,





but mediated by SS effect on budburst date in *Qrob* (Fig. 5). The proportion of variance explained by SEM models was lower for $D_h$ than for budburst date. Similarly, $R^2$ scores for EVP substantially differed between species, with values of 0.06 and 0.20 in *Qrob* and *Qpyr*, respectively (Fig. 5a, b).

EVP and $D_h$ of the current year had a strong positive influence on subsequent latewood growth in both species,
attaining a relative influence above 85 % (Fig. 6). A second group of variables included in the models was related to tree vigour, such as foliage density, with a positive effect and a relative importance of 40–60 %; and length of the growing season, with a stronger influence for *Qrob* (40 %) than for *Qpyr* (31 %). As expected, a higher foliage density together with a longer growing season predicted a larger latewood growth. Winter NSC accounted for a marginal weight in both species, having however a negative impact on latewood growth for *Qrob*, whereas positive for *Qpyr* (Fig. 6). Problems of collinearity
were not detected among the predictors included in the model (Supplement, Table S1).

## 4 Discussion

### 4.1 NSC allocation to xylem growth reflects contrasting stress-tolerance strategies in oaks

According to our expectations, the sub-Mediterranean *Qpyr* exhibited a higher SS-to-starch ratio at the onset of dormancy than the temperate *Qrob*. Sugars are involved in the osmotic protection against freezing damage (Améglio et al., 2004),
whereby cold acclimation requires the accumulation of high symplastic SS concentration in winter (Morin et al., 2007). This process is influenced by the timing of leaf shedding, which occurred earlier for *Qpyr* than for *Qrob*, reflecting the stronger cold tolerance of the former species. Additionally, SS content is maintained over a certain threshold to mitigate detrimental effects of eventual extreme events in long-lived trees (Sala et al., 2012). This may be of high relevance in the Mediterranean area, where fire and drought-induced defoliation are frequent (Rosas et al., 2013; Camarero et al., 2015). Indeed, drought-
tolerant species are thought to increase stored NSC at the cost of reducing growth (Valladares et al., 2000; Piper, 2011). This idea is further supported by the higher SS concentrations reported at the subhumid site, which also exhibited higher starch content for *Qpyr*. Under drought, sugars contribute to prevent desicattion through osmotic regulation and cavitation repair (Salleo et al., 2009; Brodersen and McElrone, 2013; Pantin et al., 2013; Deslauriers et al., 2014), although the prevalence of vessel refilling is still under discussion (Delzon and Cochard, 2014). In this regard, it is noteworthy that *Qpyr* exhibited
larger vessels at the humid and subhumid locations, being even larger than those of *Qrob*. Trade-offs between efficiency and safety thus suggest that *Qpyr* prompted long-distance water transport under drier conditions, but concurrently increased the risk of vessel dysfunction (Sperry et al., 1994). One possible explanation is that enhanced SS concentration might be involved in compensating hydraulic vulnerability in this species, although further research on SS fractioning and longer records of vessel size mesurements may be required to confirm such hypothesis.
Wood formation declined along with NSC from the subhumid toward the hyperhumid site, which is opposed to our expectations. A lower tree density at the subhumid than at the Atlantic sites probably favoured carbon uptake and growth due to more reduced inter-tree competition (Fernández-de-Uña et al., 2016). Nevertheless, strong differences between the two Atlantic sites in *Qpyr* might be attributed to contrasting moisture, instead to the similar tree density. Indeed, *Qpyr* trees exhibited sparser foliage and more severe powdery mildew infestation at the hyperhumid site, which may reduce carbon
uptake and growth (Améglio et al., 2001, Martínez-Vilalta, 2014, Camarero et al., 2015). On the other hand, soil water excess in winter could exacerbate carbon consumption associated to fermentation processes and root anaerobic stress (Ferner et al., 2012). Furthermore, the growing season was shorter at the hyperhumid and humid locations, probably restricting the carbon gain (Morecroft et al., 2003), as well as xylem formation (Rathgeber et al., 2011). Interestingly, growth decline and tree dieback were recently reported in oaks suffering from both high competition levels and water excess after extremely
rainy periods (Rozas and García-González, 2012). In contrast, more immediate effects of water shortage on stem growth than on photosynthesis likely favour carbohydrate accumulation in summer (Sala et al., 2012, Lempereur et al., 2015).



EVP and latewood width were generally lower for *Qpyr* than for *Qrob*, but such differences were striking at the subhumid site in spite of the more Mediterranean climatic conditions. This apparently contradictory result supports that *Qpyr* is more conservative than *Qrob* in allocating NSC to wood production. This is also consistent with results from Rodríguez-Calcerrada et al. (2008), who suggested that temperate oaks are more competitive than sub-Mediterranean ones. Hence, the carbon saving strategy here suggested for *Qpyr* could entail a high opportunity cost in favourable environments (Chapin et al., 1990), where coexisting *Qrob* probably outcompetes *Qpyr*. In this regard, Grossiord et al. (2014) reported that temperate sessile oaks exerted a negative effect on coexisting Turkey oaks, which experienced a reduction of their transpiration fluxes as a result of increasing water stress.

### 4.2 Dependencies among NSC, phenology, and earlywood vessels

Our SEM model confirmed the hypothesized functional relationships among earlywood anatomy, NSC, and date of budburst. Tree size had a positive direct effect on the hydraulic capacity (i.e. larger vessels), and SS content at dormancy. Wider vessels would be formed at the tree base of large trees to counteract the increasing hydraulic resistance with height (Petit et al., 2008). This was corroborated by the significant correlation between stem diameter and tree height in both studied species. Vessels of large diameter may boost carbohydrate uptake under a high evaporative demand (Fichot et al., 2009), which is a consequence of the enhanced water transport capacity (Meinzer et al., 2005). Alternatively, since large vessels are thought to be more vulnerable to cavitation (Sperry et al., 1994), higher SS concentrations may be required in the sapwood of trees bearing wider vessels to maintain long-distance water transport (Brodersen and McElrone, 2013).

Trees having a higher SS concentration in the stem showed earlier budburst the following spring, as reported in poplar (Park et al., 2009). This is related to the fact that carbohydrate influx from vessels may promote bud development (Maurel et al., 2004; Bonhomme et al., 2010). In this regard, Lacointe et al. (2004) suggested that carbon transport during the dormant period relies on xylem vessels. In addition, xylem sap osmolarity plays a role in the generation of stem pressure, which is needed to reverse winter embolisms in early spring (Améglio et al., 2001). It is noteworthy that vessels of previous years may be responsible for carrying water over quiescence because first-formed earlywood vessels are not functional at least until budburst (Pérez-de-Lis et al., 2016). Thereby, reduced ability to repair winter embolism could affect negatively the supply of water to swelling buds in those trees showing lower SS content in the xylem sap (Améglio et al., 2001).

In *Qrob*, the observed positive effect of SS concentration on EVP was mediated by the timing of budburst. The observation that budburst coincides with the onset of vessel maturation in the stem suggests that early-flushing trees had an advanced onset of earlywood formation (Pérez-de-Lis et al., 2016). Therefore, larger EVP in early-flushing trees could result from the longer period of earlywood formation, which is underpinned by that greater EVP values measured at the location with earlier budburst. Yet, the impact of SS concentration on EVP was more pronounced for *Qpyr*, as demonstrated the fact that EVP was in tune with SS concentration. Presumably, high overwintering sugar levels in sapwood somehow increased energy and materials as well as water supplied to growing tissues in spring, even though starch mobilization may be initiated at that time (Améglio et al., 2001). Furthermore, sugars are elicitors of auxin biosynthesis and distribution (Lilley et al., 2012; Sairanen et al., 2012), as well as growth promoters (Stewart et al., 2011). Although relations between carbon allocation to storage and growth are complex, and mainly related to the activity of carbon sinks (Lempereur et al., 2015), a growing body of literature suggests that NSC availability is involved in growth regulation (Pantin et al., 2013; Dietze et al., 2014; Guillemot et al., 2015). This might be particularly true for earlywood given its reliance on winter NSC reserves (Skomarkova et al., 2006). This idea agrees with the direct association between tree vigour, NSC pool, and growth found in multiple species (Sundberg et al., 1993; Deslauriers et al., 2009; Carbone et al., 2013), but also with the observed positive effect of $CO_2$ fertilization on tree growth (Nissinen et al., 2016).

Feedbacks between SS content and EVP differed between species, which may be attributed to their contrasting stress tolerance (Guillemot et al., 2015). Our results therefore suggest that the more drought-resistant *Qpyr* limits construction



costs under favourable conditions in spring if NSC levels decrease, which conveys its more conservative carbon use strategy. This result supports that storage is an actively competing sink, rather than a passive compartment (Dietze et al., 2014). Deslauriers et al. (2014) attributed drought-induced growth decline in black spruce to the increasing demand of NSC for osmotic purposes, together with dehydration effects on cell turgor. In the same line, Anderegg et al. (2013) suggested that

reduced carbon uptake under drought can impair growth in subsequent years. Yet, we noted increasing NSC concentrations and higher EVP under drier conditions, whilst sapwood starch concentrations scarcely affected growth. EVP was probably affected by the starch breakdown rate at the onset of dormancy, rather than by drought-induced changes in the reserve pool the previous summer. Our results hint that earlywood vessel anatomy and winter temperature connections reported by dendrochronological studies (Gea-Izquierdo et al., 2012) could be attributed to thermal-induced changes in carbon

partitioning during cold hardening (Améglio et al., 2004; Morin et al., 2007). However, not all the starch that is contained in sapwood is readily accessible (Sala et al., 2012). Thereby, the assessment of mean starch concentrations in sapwood probably failed to reflect their actual availability.

### 4.3 Earlywood anatomy is a predictor of latewood growth

The most influential predictors driving latewood growth did not differ between *Qrob* and *Qpyr*, suggesting common

underlying mechanisms for both species. Latewood width was influenced by earlywood properties within the same tree ring, and, to a lesser extent, by foliage density and phenology. This confirms the positive impact of enhanced water transport capacity on xylem growth (Fichot et al., 2009), which is largely related to both conduit size (Sperry et al., 1994), and total conductive area (Meinzer et al., 2005). Despite the higher construction costs, abundant earlywood vessels of distinct size could be useful to mitigate the physiological effects of drought, because still functioning conduits would serve as local water

reservoirs to recover neighbouring collapsed ones (Brodersen and McElrone, 2013). This is consistent with the higher EVP observed in *Qrob* at the subhumid site. In addition, an efficient hydraulic network may enable carbon gain to be maximized under favourable conditions (Fichot et al., 2009), protecting key processes in which sugars are involved, such as osmotic regulation (Sala et al., 2012; Deslauriers et al., 2014) and embolism repair (Salleo et al., 2009).

        Latewood growth was also higher in trees exhibiting high foliage density, which is consistent with the reduced carbon

reserves and radial growth found in defoliated trees for evergreen oaks (Rosas et al., 2013; Camarero et al., 2015), or walnut trees (Améglio et al., 2001). The lower importance of this factor on latewood growth could also be related to the high overall foliage density levels at the study sites. Trees having earlier budburst showed delayed leaf shedding and thus longer growing season, as found in previous studies on cambial activity (Deslauriers et al., 2009; Rathgeber et al., 2011). Duration of the growing season had limited influence on latewood width, suggesting that cambial activity rates were more influential on

xylem production than phenology (Deslauriers et al., 2009; Rathgeber et al., 2011).

### 5 Conclusions

In this study, non-structural carbohydrates in sapwood, wood anatomy, and leaf phenology were comprehensively addressed in two ring-porous species during one year, along a broad geographical range in NW Iberian Peninsula. Our results reveal that feedbacks between earlywood vessels and soluble sugars involve changes in wood production. Earlywood vessel

formation in *Q. pyrenaica* showed a tighter control by soluble sugar content than in *Q. robur*, suggesting a more conservative carbon use strategy in the former species. These lines of evidence support that non-structural carbohydrates play a role in the acquisition of resistance to cope with harsh environmental conditions in the sub-Mediterranean area. This study is a first attempt to unravel the interactions between non-structural carbohydrates, wood anatomy, and phenology in ring-porous oaks. We acknowledge the need for further research comprising a longer time span, soluble sugar fractioning,

additional tree compartments such as branches and roots, and a comprehensive dataset on cambial phenology instead of isolated leaf phenophases. However, this study hints the existence of stable functional interactions between sapwood



carbohydrate levels, xylem anatomy, and phenology in ring-porous oaks. In the light of these results, we suggest that *Q. pyrenaica*, and to a lesser extent *Q. robur*, might mitigate increasing hydraulic vulnerability under climate warming by prioritizing carbon accumulation over growth. Nevertheless, such mechanism would impose additional limitations for secondary growth if adverse climate episodes become more frequent in future decades.

*Author contribution* IGG, JMO, VR conceived and designed the experiment. IGG, VR, GP conducted fieldwork. GP performed sample processing and data collection. JMO and GP executed model calculation. GP prepared the manuscript. IGG, JMO, VR provided editorial advice.

*Acknowledgements* The authors are grateful to G. Juste and E. Marcos for laboratory technical assistance, to A. García-Cervigón for advice on SEM models, and to M. Souto and G. Guada for their contribution in sample collection. We also thank to L. Costa and C. Franco from the Fragas do Eume Natural Park, Forest Service of Xunta de Galicia, and MVMC of Moreiras for facilitating fieldwork. This work was supported by the Spanish Ministry of Economy and Competitiveness [Research Projects BFU-21451 and CGL2012-34209] with European Regional Development Fund, and by Xunta de Galicia
[Research Project 10MDS291009PR]. G. Pérez-de-Lis received a PhD FPU-ME grant [grant number AP2010-4911] funded by the Spanish Ministry of Education. This work was partially inspired within the FPS COST Action FP1106 – STReESS and Ecometas net CGL2014-53840REDT. The authors declare no conflict of interests of any kind.

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



**Table 1.** Climatic information of the study sites in 2012 and 2013.

| Site | P (mm) | Rainy days | Tm (ºC) | Tmax (ºC) | Tmin (ºC) |
|---|---|---|---|---|---|
| 2012 | | | | | |
| Hyperhumid | 1346.8 | 210 | 11.7 | 16.4 | 7.8 |
| Humid | 858.3 | 169 | 10.4 | 14.9 | 7.1 |
| Subhumid | 555.0 | 172 | 12.4 | 16.8 | 8.5 |
| 2013 | | | | | |
| Hyperhumid | 1979.0 | 225 | 11.6 | 15.8 | 8.0 |
| Humid | 1351.6 | 190 | 10.2 | 14.2 | 7.2 |
| Subhumid | 856.3 | 168 | 12.3 | 16.4 | 8.8 |

*P* mean precipitation, *Tm* mean temperature, *Tmax* mean maximum temperature, *Tmin* mean minimum temperature.



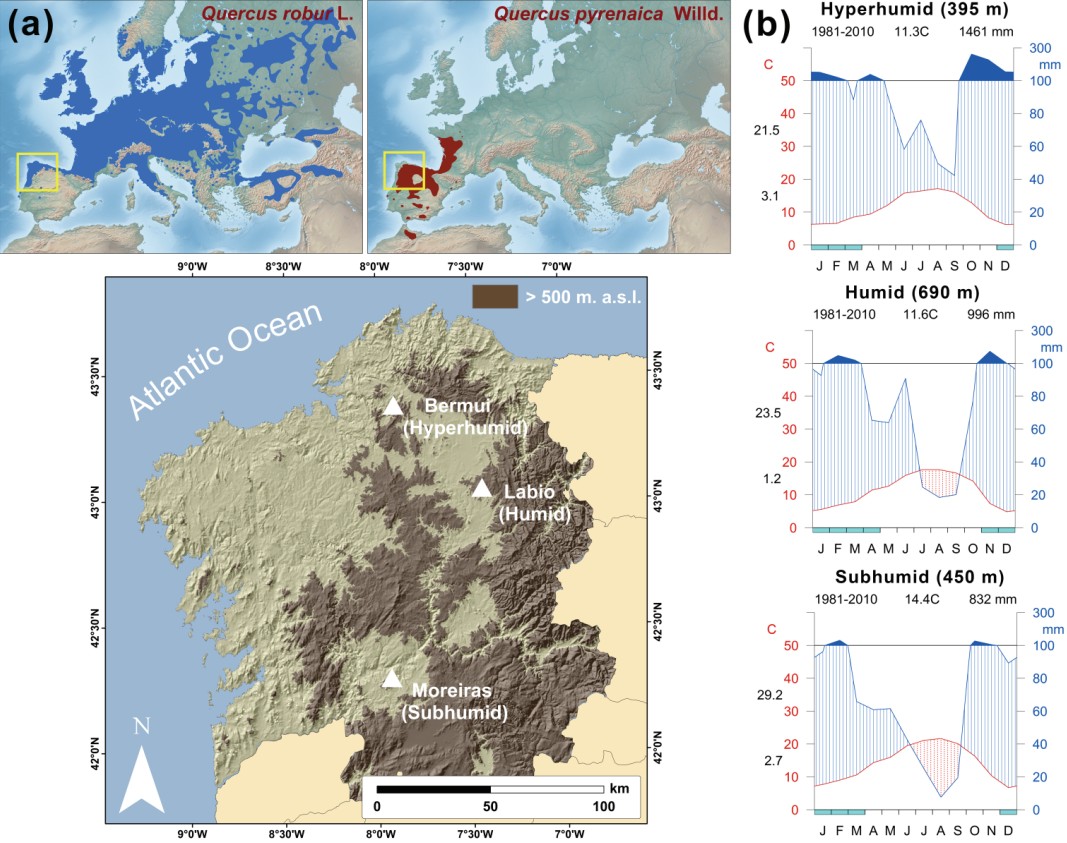

**Figure 1: (a)** Location of the study sites in NW Iberian Peninsula and distribution range of *Quercus robur* and *Quercus pyrenaica* (base map: http://www.euforgen.org). **(b)** Climatic diagrams of the sites including site altitude in m a.s.l, mean annual temperature, and total annual precipitation for the specified period.




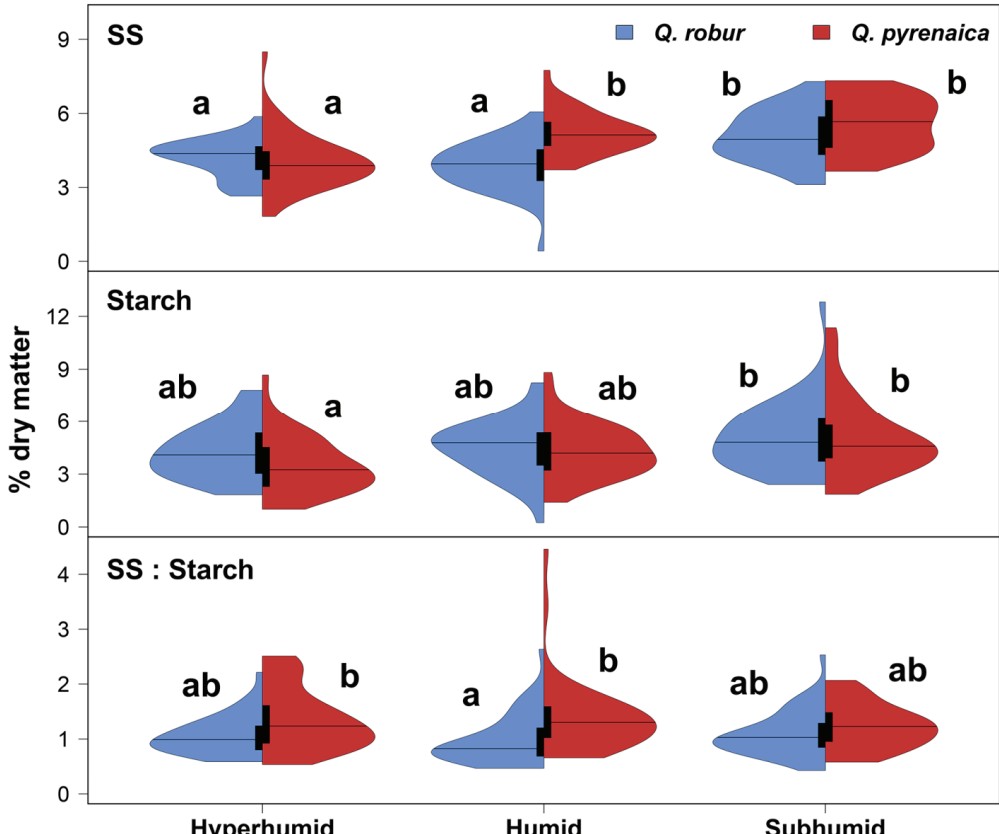

**Figure 2: Distribution of soluble sugars (SS), starch concentrations, and SS-to-starch ratio, for *Quercus robur* and *Q. pyrenaica* at the three study sites. Horizontal lines represent the median, and black boxplots show the extent of 25th and 75th percentiles. Lower case letters indicate statistically significant differences along the gradient according to multiple pairwise comparisons.**




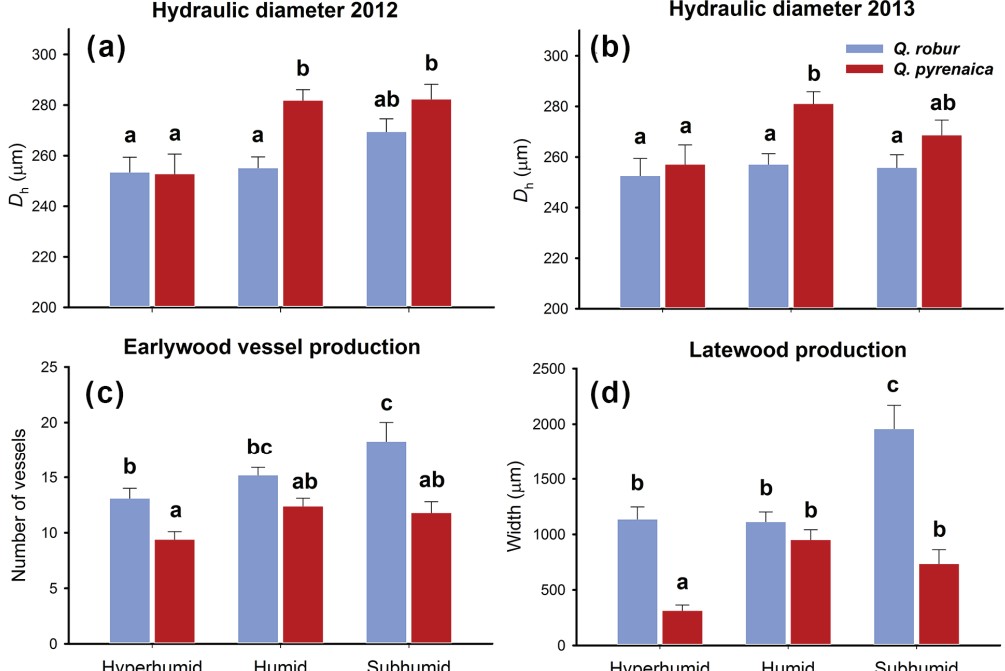

**Figure 3: Mean values and SE of (a) hydraulic diameter in 2012, (b) hydraulic diameter in 2013, (c) earlywood vessel production, and (d) latewood production in 2013 for *Quercus robur* and *Q. pyrenaica*. Lower case letters indicate statistically significant differences along the gradient according to multiple pairwise comparisons.**



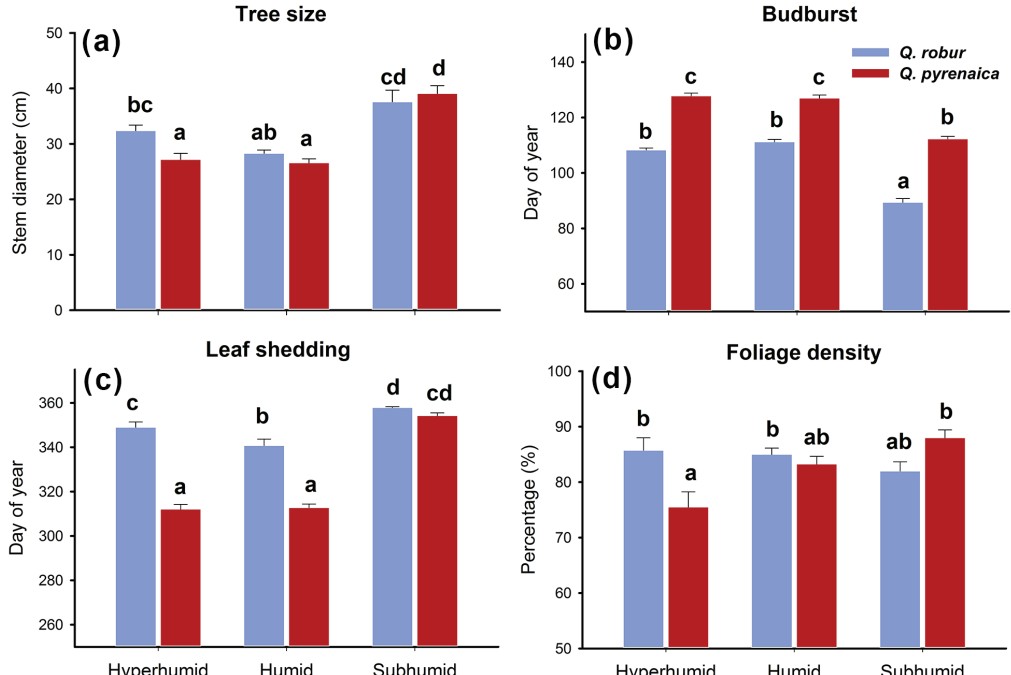

**Figure 4: Mean values and SE of (a) stem diameter, (b) date of budburst, (c) date of leaf shedding, and (d) foliage density in 2013 for *Quercus robur* and *Q. pyrenaica*. Lower case letters indicate statistically significant differences along the gradient according to Multiple Pairwise Comparisons.**




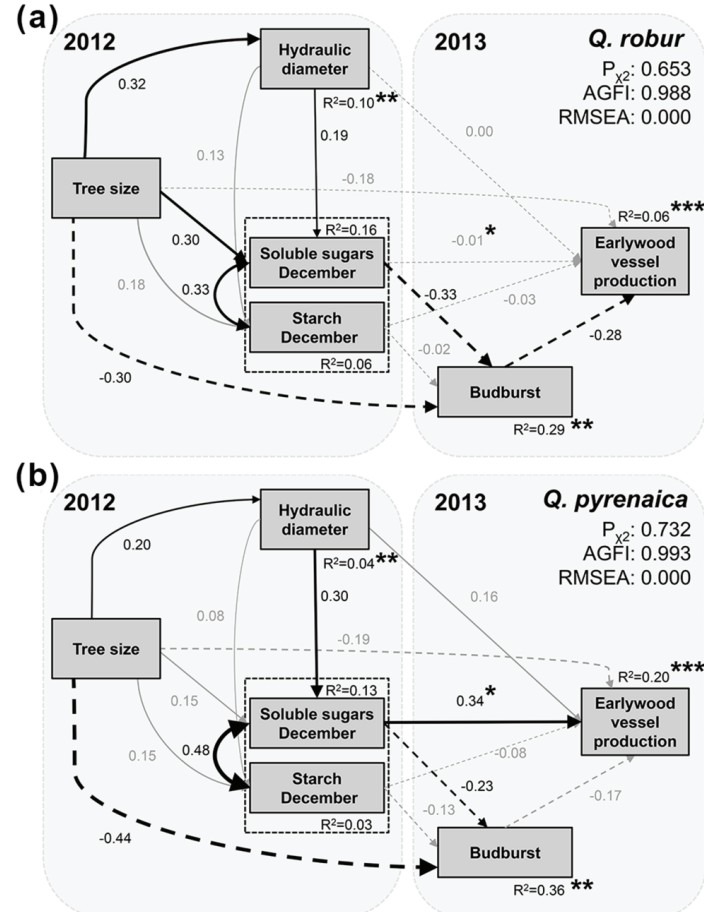

**Figure 5: Structural equation models fitted for (a)** *Quercus robur* **and (b)** *Q. pyrenaica*. **Variables of the conceptual model are tree size (stem diameter), hydraulic diameter in 2012, soluble sugars and starch concentrations in December 2012, date of budburst, and earlywood growth (number of vessels) in 2013. Explained deviances of endogenous variables are shown near the boxes. Black solid (positive effects) and dashed (negative effects) arrows denote significant relations, while non-significant relations are shown as grey coefficients and arrows. The Chi-square test, the adjusted goodness of model fit index (AGFI), and the root mean square error of approximation (RMSEA) are shown for each model. Asterisks indicate paths or error values significantly different between the models of both species. \*\*\*** $P \leq 0.001$**, \*\*** $P \leq 0.01$**, and \*** $P \leq 0.05$**.**





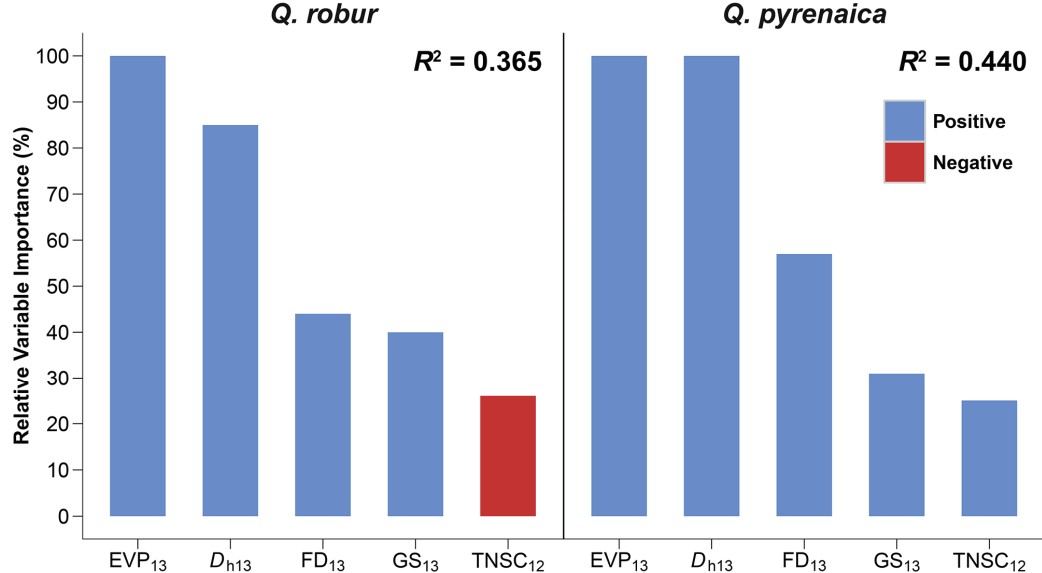

**Figure 6: Relative importance of the variables driving latewood production in 2013, expressed as percentage. Location effect was included as random factor in the model. $NSC_{12}$ is total non-structural carbohydrates in December 2012, $D_{h13}$ is hydraulic diameter in 2013, $EVP_{13}$ is earlywood vessel production in 2013, $GS_{13}$ is length of the growing season in 2013, and $FD_{13}$ is foliage density in 2013. Different colours of bars denote variables with either a positive or negative effect. We provide the coefficient of determination ($R^2$) of the full model.**