# Peer review of "Feedbacks between earlywood anatomy and non-structural carbohydrates affect spring phenology and wood production in ring-porous oaks"

_Biogeosciences, 2016_

## Referee Comment (RC1) · Anonymous Referee #1 · 24 Jun 2016

Pérez-de-Lis et al put forth a commendable study on the correlations between NSC reserves and tree physiology, particularly xylem structure and function. This topic first within the scope of BG and presents some novel concepts. While the authors formulate conclusions to hypothesis put forth, there appear to be significant limitations in their support for hypotheses (i) and (ii) (p2, lines 38-40). The description of the experiments is adequate with some significant issues (see below). Proper credit is given to related work. The title is appropriate for the research. I urge the authors to reduce the discussion in a effort to strengthen support for their findings.

[Figure]

Regarding methods, there is no indication on how DBH or tree height was measured. Correlations between DBH and tree height was not described statistically. In the methods section, the authors state that 40 trees per species were selected, but in the methods or results sections, there is no indication on sample size for NSC or anatomical measurements. Can the reader assume n=40 for all comparisons?

Authors fail to account for age of the trees when estimating growth rate by measurement of DBH. Authors state that DBH scales with tree height, but no statistics are offered to justify such allometric scaling. Thus, I find it problematic to use only DBH as an indication of tree size because of the disregard to growth rates. Furthermore, calculations of BAI would be useful in correlating NSC reserves with growth rate and subsequent parameters such as EVP, bud break, latewood production, etc...

For the results section, comparisons are not adequately stated. Examples include: page 5 line8 – is this comparison on NSCs combined across all sites; page 5 line 11 – the figure implies no significant difference between species at hyperhumid in the SS:starch ratio; page 5 line 12 – NSC (being a total of SS and starch) is not indicated in the figure and is this a comparison of species across all sites?; page 5 line 17 – Fig is not referenced correctly; page 5 line 18 – fig implies that hyperhumid does not differ from subhumid; page 5 line 19 – it is not clear which species are being referred to here; page 5 line 26 – clarification is needed here as to what species is being referred to for the budburst range, furthermore, are these comparisons referring to min/max, as the figure implies means of only end of March ($\sim$90 days) to early May ($\sim$130); page 5 line 27 – clarify what is being compared here; page 5 line 29 – this correlation does not appear to be consistent across all sites; page 5 line 34 – what is "ALT" referring to?

While the authors acknowledge limitations of this study, in particular the need to include more tissue types for NSC analyses and subsequent comparisons, using only stemwood NSC reserves as a proxy for hypotheses put forth in this research is possibly flawed. A primary concern is that stores of NSC reserves in the root system could have a huge impact on growth, budburst, etc, and cannot be ignored. Such analyses

would need to be conducted in order to present this research as acceptable.

---

## Author Comment (AC1) · 11 Jul 2016

REVIEWER: Pérez-de-Lis et al put forth a commendable study on the correlations between NSC reserves and tree physiology, particularly xylem structure and function. This topic first within the scope of BG and presents some novel concepts. While the authors formulate conclusions to hypothesis put forth, there appear to be significant limitations in their support for hypotheses (i) and (ii) (p2, lines 38-40). The description of the experiments is adequate with some significant issues (see below). Proper credit is given to related work. The title is appropriate for the research.

[Figure]

ANSWER: We strongly appreciate the comments by the referee, which give us the chance to improve the quality of our work.

REVIEWER: I urge the authors to reduce the discussion in a effort to strengthen support for their findings.

ANSWER: We thank this comment. Indeed, sub-section 4.1 "NSC allocation to xylem growth reflects contrasting stress-tolerance strategies in oaks" can be reduced by re-ordering some of the ideas presented there. We will perform these modifications accordingly upon receipt of the other reviewers' suggestions.

REVIEWER: Regarding methods, there is no indication on how DBH or tree height was measured.

ANSWER: This information was missing in the former version of the manuscript. Stem diameter was measured by using a diameter tape, while we used a Blume-Leiss hypsometer to estimate tree height. This information is properly included in the Materials and Methods section of the revised version.

REVIEWER: Correlations between DBH and tree height was not described statistically.

ANSWER: We performed Pearson's correlations (two-tailed test of significance and 0.95 confidence interval). This information is included in Materials and Methods of the revised version.

REVIEWER: In the methods section, the authors state that 40 trees per species were selected, but in the methods or results sections, there is no indication on sample size for NSC or anatomical measurements. Can the reader assume n=40 for all comparisons?

ANSWER: Sample size is 240 (40 by species at each site) throughout the manuscript. At the beginning of the sub-section entitled "2.2 Phenology and NSC concentration" we stated that "40 trees per species were selected for sampling". For the sake of clarity we added the expression "in all the selected trees" in page 3 line 28 of and page 4 line1.

[Figure]

REVIEWER: Authors fail to account for age of the trees when estimating growth rate by measurement of DBH. Authors state that DBH scales with tree height, but no statistics are offered to justify such allometric scaling. Thus, I find it problematic to use only DBH as an indication of tree size because of the disregard to growth rates. Furthermore, calculations of BAI would be useful in correlating NSC reserves with growth rate and subsequent parameters such as EVP, bud break, latewood production, etc...

ANSWER: As the reviewer rightly commented, growth rates were not considered in our study. Actually, our SEM model was focused on relations between NSC, hydraulic capacity, and vessel production, rather than on growth rates. In order to avoid misunderstanding, we removed both "fast-growing" and "slow-growing" expressions from the manuscript. We also replaced "fast-growing trees" by "dominant trees" in page 2 line 3, and "than their slow-growing counterparts" was removed in line 4. Similarly, in page 4 line 23 we replaced "fast-growing trees (larger stem diameter)" by "bigger trees".

The reason for us to consider "tree size" in our study was because it is assumed to affect wood anatomy, as well as to carbon economy and storage (i.e. Petit et al. 2008, New Phytol; McDowell et al. 2005, Oecol; Sala and Hoch et al. 2009, Plant, Cell Environ). In addition, large trees are frequently dominant, having a different timing of xylogenesis and phenology than small trees (Rathgeber et al. 2011, Ann Bot). In our view, "stem diameter" is a good indicator of tree size in this study because tree height and stem diameter were noted to be positively correlated (Page 5 line 22; but see also the response to the comment: page 5 line 29 – this correlation does not appear to be consistent across all sites). However, if the referee still judges it necessary, we will replace "tree size" by "stem diameter" in the revised version.

REVIEWER: For the results section, comparisons are not adequately stated. Examples include: page 5 line 8 – is this comparison on NSCs combined across all sites;

ANSWER: We performed comparisons across sites, indeed. We will substitute "Mean SS concentrations at the sites ranged from 3.88 to 5.08 percentage dry matter in Qrob"
by "Mean SS concentrations ranged along the gradient from 3.88 to 5.08 percentage dry matter in Qrob" (page 5 line 6).

REVIEWER: page 5 line 11 – the figure implies no significant difference between species at hyperhumid in the SS:starch ratio;

ANSWER: Thanks for the comment. In the revised version, hyperhumid location is removed from this sentence.

REVIEWER: page 5 line 12 – NSC (being a total of SS and starch) is not indicated in the figure and is this a comparison of species across all sites?

ANSWER: Actually, we compared values of both species (together) among study sites. We removed the reference to Figure 2 from this sentence.

REVIEWER: page 5 line 17 – Fig is not referenced correctly;

ANSWER: The reference to Figure 3a,b should have appeared in the following sentence. In the revised version, it is included in the proper position.

REVIEWER: page 5 line 18 – fig implies that hyperhumid does not differ from subhumid;

ANSWER: Actually, differences between the hyperhumid and the subhumid locations are significant in 2012, but not in 2013. In the revised version, we rewrote this phrase as follows: "In Qpyr, Dh at the hyperhumid site was substantially lower than at the humid and subhumid locations in 2012, but only than at the humid site in 2013. By contrast, much less variation was found among sites for Qrob".

REVIEWER: page 5 line 19 – it is not clear which species are being referred to here;

ANSWER: We rewrote this phrase in a clearer form as follows: "EVP values decreased from the subhumid to the hyperhumid location in Qrob (F = 5.55, P = 0.005; Fig. 3c) and Qpyr (F = 4.12, P = 0.019). Trees of both species exhibited wider latewood at the subhumid than at the hyperhumid site (Qrob F = 11.88, Qpyr F= 14.25, P < 0.001, Fig.

3d)".

REVIEWER: page 5 line 26 – clarification is needed here as to what species is being referred to for the budburst range, furthermore, are these comparisons referring to min/max, as the figure implies means of only end of March (90 days) to early May (130);

ANSWER: Dates provided are those corresponding to min/max values. In the revised version we specify that such information is referred to both species.

REVIEWER: page 5 line 27 – clarify what is being compared here;

ANSWER: We rewrote this sentence as follows: "Budburst at the hyperhumid and humid sites occurred synchronously (Qrob t=1.92, Qpyr t=–0.54, P>0.05), but later than at the subhumid location (Qrob F=128.45, Qpyr F=79.49, P<0.001)".

REVIEWER: page 5 line 29 – this correlation does not appear to be consistent across all sites;

ANSWER: Correlation coefficients at the former version of the manuscript were calculated at species level, after pooling the data of the three locations. Actually, stem diameter and tree height are strongly correlated at the hyperhumid and subhumid sites, while correlation is not significant at the humid location (See supplement to the comment, Table 1). The low variance in both stem diameter and height values measured at this latter location, which could be attributed to low age variability, likely obscured the scaling association between these two variables (See supplement to the comment, Fig. 1). Since this study is not focused on relations between stem diameter and tree height, we propose to include this information as Supplementary material.

REVIEWER: page 5 line 34 – what is "ALT" referring to?

ANSWER: We apologize for this mistake; we put "ATL" instead of "Hyperhumid site". This error has been corrected in the revised version.

[Figure]

REVIEWER: While the authors acknowledge limitations of this study, in particular the need to include more tissue types for NSC analyses and subsequent comparisons, using only stemwood NSC reserves as a proxy for hypotheses put forth in this research is possibly flawed. A primary concern is that stores of NSC reserves in the root system could have a huge impact on growth, budburst, etc, and cannot be ignored. Such analyses would need to be conducted in order to present this research as acceptable.

ANSWER: As the referee pointed out, we are aware of the limitations of our study, which were detailed in the Conclusion section. Despite this, we think that the work presented in our manuscript is novel and supposes an incremental advance in this topic that may be interesting for a broad audience. In this regard, we would like to point out that several anatomical and phenological parameters (vessel size and number, ring width, budburst dates, stem diameter) were assessed simultaneously with carbohydrate reserves for the first time in ring-porous species. In addition, our data set is large, involving two species and three sites along a rainfall gradient (and 240 trees).

Carbon reserves in deciduous trees are mostly stored in the stem, as well as in coarse roots and branches (Barbaroux et al. 2003, New Phyt). Probably, additional NSC measurements in roots and branches would have been essential if our objective were to analyze changes in the total NSC pool, as it has been done in studies testing the carbon starvation hypothesis (i.e. Anderegg and Anderegg 2013, Tree Phys; Galvez et al. 2013, New Phyt; Hartmann et al. 2013, Func Ecol); or if we were interested in analyzing carbon fluxes among different tree compartments (Regier et al. 2010, Tree Phys). However, although we recognize that NSC measurements in roots would have been valuable in this study, our objective was to disentangle the interaction between xylem growth and short-distance NSC content, rather than focusing on the differences in NSC storage patterns among populations or species.

According to Steppe et al. (2015, Trends Plant Sci), a mechanism of xylem growth is dependent on the incorporation of carbon resources, and thus requires information on sugar concentrations in the stem. In this regard, a number of recently published studies

analyze the interplay between carbohydrate content and xylem growth by sampling the cambial zone (Deslauriers et al. 2009, Tree Phys; Deslauriers et al. 2014, Ann Bot), stem sapwood (Galiano et al. 2011, New Phyt; El Zein et al. 2011, Tree Phys; Oberhuber et al. 2011, Can J For Res; Carbone et al. 2013, New Phyt), or several above ground compartments (Sala and Hoch et al. 2009, Plant, Cell Environ; Fajardo et al. 2012, New Phyt; Saffell et al. 2014, Tree Phys). It is interesting to note that NSC translocation through the different plant compartments may be strongly reduced during dormancy if phloem becomes non functional, as suggested by Lacointe et al. (2004, Plant, Cell Environ) for the deciduous walnut. Therefore, the contribution of local stem carbon reserves might probably be considerable in fuelling xylem growth before the sink-to-source transition of leaves (Begum et al. 2010, Ann Bot). This is related to the fact that initiation of cambial divisions in roots and stem precedes budburst in deciduous oaks and is consistent with the strong decline in stem NSC concentration frequently reported in spring (Barbaroux and Bréda, 2002, Tree Phys; El Zein et al. 2011, Tree Phys). For these reasons, we think that relations between stem sapwood NSC content and earlywood growth dynamics are especially relevant.

We agree that hypotheses in page 2 can be more accurately defined by specifying that interpretations are only referred to stem sapwood. Thus, in the new version, we rephrased the sentence in line 35, page 2, as follows: "In this regard, we hypothesized that the more-drought tolerant Q. pyrenaica will have larger stem sapwood NSC concentration but lower wood production than Q. robur". Likewise, line 39, page 2, we specified more clearly: "(i) tree size influences esrlywood vessel diameter, which in turn affect NSC content in the stem; (ii) higher sapwood SS content in winter predisposes trees to advance growth resumption in spring, thereby fostering earlywood production [. . .]".

Please also note the supplement to this comment:
http://www.biogeosciences-discuss.net/bg-2016-227/bg-2016-227-AC1-supplement.pdf

[Figure]

[Figure]

**Supplement:**

Table 1: Correlation coefficients (*P*–values) between stem diameter and tree height.

|  | Hyperhumid site | Humid site | Subhumid site |
|---|---|---|---|
| *Q. robur* | 0.473 (*P*=0.002) | 0.160 (*P*=0.323) | 0.685 (*P*<0.001) |
| *Q. pyrenaica* | 0.581 (*P*<0.001) | 0.175 (*P*=0.279) | 0.461 (*P*=0.003) |

Figure 1: Stem diameter versus tree height (blue dots) at the hyperhumid, humid, and subhumid sites for *Q. robur* and *Q. pyrenaica*. Regression lines are shown in red.

[Figure]

---

## Referee Comment (RC2) · Anonymous Referee #2 · 1 Aug 2016

Overall, this was an interesting and useful contribution to the ongoing discussion about the roles of NSCs and plant hydraulics on tree phenology, growth, and survival. In this paper the authors studied two congenator oaks of that contrast in their ecological strategies to compare the impacts of winter NSC storage, hydraulic diameter, and budburst on earlywood vessel production (EVP) and the subsequent impacts of EVP, hydraulic diameter, foliar density, growing season length, and NSC on latewood production. Species were evaluated at three sites that form a moisture gradient in northwest Spain.

[Figure]

This paper was generally well written and well cited and most of my concerns are moderate and should not change the overall results.

Page 2, Lines 22-24: Here you describe one of your study species, but you fail to describe the other. I know Q. robur is more common, but not all your readers will be familiar with its ecology.

Page 3, Line 21: No description is given as to HOW the trees were selected. In particular, I have no idea if the authors put out plots of some standard design, picked 'representative' trees, or picked the 40 biggest, healthiest trees they could find. No description is given of the size threshold or other criteria for inclusion (we could in theory be comparing a sapling at one site to a 100cm DBH tree at another). Unfortunately, ample evidence exists to show that trees and locations chosen subjectively to be 'normal' or 'representative' tend to be far better off than random, which unfortunately would cause all of the ANOVA-based comparative analyses to fall into question and require very careful interpretation of the regression-based analyses. I think in any revision the authors need to provide considerable more information about sampling and the editor should pay careful attention this information in assessing the validity of the work. For the remainder of the review I'm going to assume the sampling was done correctly (randomized locations, randomized trees within location).

Page 3, Line 32: How was sapwood area determined?

Page 4, Line 37: I'm going to assume growing season length is an individual-level measure and not a site-level measure (as is commonly done), otherwise this effect is confounded with the site random effect

Page 5, Lines 1-2: Here you're talking about averaging over a set of models, but in the paragraph above you only describe a single model. Where does this other set of models come from? Why do you need another set of models? Why is the sum of Akaike weights an appropriate measure of the relative importance of a variable? This quantity is quite challenging to interpret, especially in a GLMM, and fairly unintuitive.

I'm all for sophisticated analyses when needed, but why not stick to a simpler analysis (e.g. the proportion of the variance [$R^2$] explained by each covariate), which in my mind would be much easier to interpret and a more direct measure of importance. As I tend to look at the figures before I read a paper, I'll also note that the meaning of 'relative importance' (essentially a weighted number of times that a variable was included in the model) is not clear in the figure.

Page 5, Line 8: You should report the degrees of freedom in the F test (and all other tests). If this is going to be the same for all subsequent analyses state that here at the first usage, otherwise make the df explicit for each analysis

Page 5, Line 34: Be consistent with notation. In all other places you refer to sites by their moisture status, and here you've reverted to a site code, and I'm not sure which site you're referring to

Page 5, Line 35: Were trees with powdery mildew included or excluded? Why wasn't this included as a covariate? Why is there not more in the discussion about how this could be affecting results?

Page 5, Line 37 to Page 6, Line 3: In the Results (here) and Discussion (below), I'm concerned that the authors are over-interpreting the biological significance of results that are statistically significant but have low $R^2$. Looking at Figure 5, about all I'm comfortable concluding is that SS and tree size have a negative impact on budburst in both species, and that SS had a positive impact on EVP in Q. pyrenaica. Effects in the $R^2$ of 3-6% range (Starch, Q robor EVP) don't seem worth discussing, and those in the 10-16% range (Dh, SS) should be acknowledged as weak.

Page 6, line 31: Tree density effects are speculative

---

## Author Comment (AC2) · 6 Aug 2016

REFEREE: Overall, this was an interesting and useful contribution to the ongoing discussion about the roles of NSCs and plant hydraulics on tree phenology, growth, and survival. In this paper the authors studied two congenator oaks of that contrast in their ecological strategies to compare the impacts of winter NSC storage, hydraulic diameter, and budburst on earlywood vessel production (EVP) and the subsequent impacts of EVP, hydraulic diameter, foliar density, growing season length, and NSC on latewood production. Species were evaluated at three sites that form a moisture gradient

in northwest Spain.

This paper was generally well written and well cited and most of my concerns are moderate and should not change the overall results.

ANSWER: We kindly thank the referee for taking our discussion paper under consideration, and for the helpful and constructive comments raised to improve the quality of our research. We agree with the referee that some methodological aspects need to be more clearly stated. In our opinion, we can easily address such points in a revised version of the paper. Our proposed changes are listed next to the points raised by the referee.

REFEREE: Page 2, Lines 22-24: Here you describe one of your study species, but you fail to describe the other. I know Q. robor is more common, but not all your readers will be familiar with its ecology.

ANSWER: In the revised version, this paragraph is modified as follows: "This is the case of the ring-porous oaks Quercus robur L. and Q. pyrenaica Willd., which coexist in NW Iberian Peninsula. The former is widespread in Europe, being abundant in areas with mild-oceanic climate. By contrast, Q. pyrenaica is dominant in various mountainous ranges of the sub-Mediterranean area, hence exhibiting multiple adaptations to cope with summer drought and winter frost, such as late flushing (Pérez-de-Lis et al., 2016)".

REFEREE: Page 3, Line 21: No description is given as to HOW the trees were selected. In particular, I have no idea if the authors put out plots of some standard design, picked 'representative' trees, or picked the 40 biggest, healthiest trees they could find. No description is given of the size threshold or other criteria for inclusion (we could in theory be comparing a sapling at one site to a 100cm DBH tree at another). Unfortunately, ample evidence exists to show that trees and locations chosen subjectively to be 'normal' or 'representative' tend to be far better off than random, which unfortunately would cause all of the ANOVA-based comparative analyses to fall into question

and require very careful interpretation of the regression-based analyses. I think in any revision the authors need to provide considerable more information about sampling and the editor should pay careful attention this information in assessing the validity of the work. For the remainder of the review I'm going to assume the sampling was done correctly (randomized locations, randomized trees within location).

ANSWER: The study was carried out at three sites where both study species were present. At all the sites, trees were randomly selected from those belonging to the study species, although suppressed individuals were disregarded. In the revised version, we will include a more detailed description of tree selection.

REFEREE: Page 3, Line 32: How was sapwood area determined?

ANSWER: Sapwood can easily be distinguished by colour. Heartwood in oaks is brown-coloured while sapwood has a pale tone (Figure 1, Supplement). For the sake of clarity, this information will be included in the revised version of the manuscript.

REFEREE: Page 4, Line 37: I'm going to assume growing season length is an individual-level measure and not a site-level measure (as is commonly done), otherwise this effect is confounded with the site random effect.

ANSWER: Indeed, growing season length is an individual-level parameter. In fact, in the discussion paper is said that "Leaf phenology was weekly monitored during 2013 using binoculars (10×) at ca. 10 m distance from each tree" (page 3 line 22). Yet, this sentence will be rewritten as follows: "Leaf phenology was weekly monitored for each tree during 2013 using binoculars (10×) at ca. 10 m distance"

REFEREE: Page 5, Lines 1-2: Here you're talking about averaging over a set of models, but in the paragraph above you only describe a single model. Where does this other set of models come from? Why do you need another set of models? Why is the sum of Akaike weights an appropriate measure of the relative importance of a variable? This quantity is quite challenging to interpret, especially in a GLMM, and fairly unintuitive. I'm all for sophisticated analyses when needed, but why not stick to a simpler analysis (e.g. the proportion of the variance [R2] explained by each covariate),which in my mind would be much easier to interpret and a more direct measure of importance. As I tend to look at the Figures before I read a paper, I'll also note that the meaning of 'relative importance' (essentially a weighted number of times that a variable was included in the model) is not clear in the Figure.

ANSWER: What we meant with "set of models" is that we calculated the AIC of the models containing all the possible fixed-effect combination. In the discussion paper we used an information-theoretic approach to identify the most influent fixed effects of the model. According to this procedure, models were compared using their AIC scores (the lower the AIC, the better the model fit). Hence, models were ranked and averaged in order to assess the relative weight of each variable (we averaged 95% of all the fitted models according to their AIC scores). As we mentioned in the discussion paper, this method was detailed in Burnham and Anderson (2002), and has been used in a recent paper analyzing possible limitations of carbon supply on secondary growth published in Biogeosciences (Guillemot et al. 2015).

In order to have a more confident analysis, the proportion of the variance explained by each predictor can also be calculated and provided (either in the main text or as a supplementary material). Although there are different model selection procedures, with different advantages and caveats, we think that combining the information-theoretic approach with the R2 of the covariates (as proposed by the referee) should be a good alternative to improve the robustness of our results. In addition, a table summarizing the estimates and significance for each covariate (in the full model) can be presented as supplementary material.

REFEREE: Page 5, Line 8: You should report the degrees of freedom in the F test (and all other tests). If this is going to be the same for all subsequent analyses state that here at the first usage, otherwise make the df explicit for each analysis.

ANSWER: We agree with the referee. df values will be included in the revised text.

REFEREE: Page 5, Line 34: Be consistent with notation. In all other places you refer to sites by their moisture status, and here you've reverted to a site code, and I'm not sure which site you're referring to.

ANSWER: We apologize for this mistake again; we put "ATL" instead of "Hyperhumid site". This error has been corrected in the revised version.

REFEREE: Page 5, Line 35: Were trees with powdery mildew included or excluded? Why wasn't this included as a covariate? Why is there not more in the discussion about how this could be affecting results?

ANSWER: Unfortunately, powdery mildew infestation has not been quantified. Yet, it is relevant to take into account that all the trees were more or less affected by the pest (it would not be considered as a covariate, but as a part of the site effect). This is a very frequent disease at oak forests in the study region, but their effects during the humid spring of 2013 were higher than usual at the hyperhumid location. Thus, we decided to provide this detail, which probably contributed to impair xylem growth at this site.

Actually, the possible effect of powdery mildew infestation was mentioned in the Discussion (page 6 line 34). Since we did not perform any measurement, we marginally commented this issue.

REFEREE: Page 5, Line 37 to Page 6, Line 3: In the Results (here) and Discussion (below), I'm concerned that the authors are over-interpreting the biological significance of results that are statistically significant but have low R2. Looking at Figure 5, about all I'm comfortable concluding is that SS and tree size have a negative impact on budburst in both species, and that SS had a positive impact on EVP in Q. pyrenaica. Effects in the R2 of 3-6% range (Starch, Q robor EVP) don't seem worth discussing, and those in the 10-16% range (Dh, SS) should be acknowledged as weak.

ANSWER: Thanks for the comment. In the revised version, we will include a more

careful interpretation of these significant relationships accounting for a low variance in the observed parameters.

REFEREE: Page 6, line 31: Tree density effects are speculative

ANSWER: This idea was based on differences in stand tree density (reported in Pag 3 line 12) and basal area (according to stem diameter measurements) among locations. We acknowledge that direct measurements on tree competition were not carried out, however recent work modelled a strong effect of competition in Quercus pyrenaica secondary growth (Fernández-de-Uña et al., 2016). The sentence in Page 6, line 31, will be rewritten as follows: "One explanation could be that lower tree density at the subhumid site might be associated to a lower inter-tree competition, which is assumed to favour both carbon uptake and xylem growth (Fernández-de-Uña et al., 2016)".

Please also note the supplement to this comment:
http://www.biogeosciences-discuss.net/bg-2016-227/bg-2016-227-AC2-supplement.pdf

**Supplement:**

Figure 1: Transverse section of a wood core of *Q. pyrenaica* showing the colour boundary between heartwood and sapwood.

[Figure]

---

## Author Response (AR1)

We kindly thank the referees for taking our discussion paper under consideration and for their helpful and constructive comments. We performed a thorough revision of the manuscript by taking into account all issues pointed out by the referees, as well as their suggestions to improve methodology, statistical analyses, and the discussion sections. Our proposed changes are listed next to the points raised by the referees. We also added in the Acknowledgements section a project reference number (GRC2015/008) that had not been included in the former version.

*REFEREE#1*

**Pérez-de-Lis et al put forth a commendable study on the correlations between NSC reserves and tree physiology, particularly xylem structure and function. This topic first within the scope of BG and presents some novel concepts. While the authors formulate conclusions to hypothesis put forth, there appear to be significant limitations in their support for hypotheses (i) and (ii) (p2, lines 38-40). The description of the experiments is adequate with some significant issues (see below). Proper credit is given to related work. The title is appropriate for the research.**

Answer: Hypotheses were more accurately defined in the revised manuscript. In this regard, we included that our interpretations were only referred to stem sapwood, rather than to the whole-tree NSC pool. For further detail, see the response to the final comment of referee#1.

**I urge the authors to reduce the discussion in a effort to strengthen support for their findings.**

Answer: We are grateful for this comment. The discussion of our results was carefully revised. Reiterative ideas were removed and the remaining text reorganized within the sub-sections. In addition, we changed the order of the hypotheses in the Introduction (page 2, lines 36-38), and modified several sentences in the summary in order to clarify key ideas of our study.

**Regarding methods, there is no indication on how DBH or tree height was measured.**

Answer: This information was missing in the former version of the manuscript. Stem diameter was measured by using a diameter tape, while we used a Blume-Leiss hypsometer to estimate tree height. It has been properly included in the Materials and Methods section of the revised version (page 3, lines 23-24).

**Correlations between DBH and tree height was not described statistically.**

Answer: We performed Pearson's correlations (two-tailed test of significance and 95% confidence interval). This information is now included in Materials and Methods (page 4, lines 18-19).

**In the methods section, the authors state that 40 trees per species were selected, but in the methods or results sections, there is no indication on sample size for NSC or anatomical measurements. Can the reader assume n=40 for all comparisons?**

Answer: Sample size is 240 (40 by species (2) at each site (3)) throughout the manuscript. For the sake of clarity we clarified that NSC, phenological and anatomical measurements were performed "from all the selected trees" in page 3, line 23 and page 4, line 4. Moreover, we provided sample size for each species in Table 2 and Figures 3-7 captions.

**Authors fail to account for age of the trees when estimating growth rate by measurement of DBH. Authors state that DBH scales with tree height, but no statistics are offered to justify such allometric scaling. Thus, I find it problematic to use only**

DBH as an indication of tree size because of the disregard to growth rates. Furthermore, calculations of BAI would be useful in correlating NSC reserves with growth rate and subsequent parameters such as EVP, bud break, latewood production, etc...

Answer: As the reviewer rightly commented, growth rates were not considered in our study. Actually, our SEM model was focused on relations between NSC, hydraulic capacity, and vessel production, rather than on growth rates. In order to avoid misunderstanding, we removed both "fast-growing" and "slow-growing" expressions from the manuscript. Hence, we replaced "fast-growing trees" by "dominant trees" in page 2 line 4, and "than their slow-growing counterparts" was removed. Similarly, in page 4 line 26 we replaced "fast-growing trees (larger stem diameter)" by "bigger trees".

The reason for us to consider "tree size" in our study was because it is assumed to affect wood anatomy, as well as to carbon economy and storage (i.e. Petit et al. 2008, *New Phytol*; McDowell et al. 2005, *Oecol*; Sala and Hoch et al. 2009, *Plant, Cell & Environ*). In addition, large trees are frequently dominant, having a different timing of xylogenesis and phenology than small trees (Rathgeber et al. 2011, *Ann Bot*). In our view, "stem diameter" is a good indicator of tree size in this study because tree height and stem diameter were positively correlated (page 5 lines 33-35 of the revised manuscript). We replaced, however, "tree size" by "stem diameter" in Figs. 2, 5a and 6 in order to avoid confusion.

**For the results section, comparisons are not adequately stated. Examples include: page 5 line 8 – is this comparison on NSCs combined across all sites;**

Answer: We performed comparisons across sites, indeed. We substituted "Mean SS concentrations at the sites ranged from 3.88 to 5.08 % dry matter in *Qrob*" by "Mean SS concentrations along the gradient ranged from 3.88 to 5.08 % dry matter in *Qrob*" (page 5 line 17).

**page 5 line 11 – the figure implies no significant difference between species at hyperhumid in the SS:starch ratio;**

Answer: Thanks for the comment. In the revised version, hyperhumid location was removed from this sentence.

**page 5 line 12 – NSC (being a total of SS and starch) is not indicated in the figure and is this a comparison of species across all sites?**

Answer: Actually, we intended to compare mean values of both species together among study sites. However, we removed this comparison regarding total NSC from the manuscript because it was reiterative and confusing.

**page 5 line 17 – Fig is not referenced correctly;**

Answer: The reference to Figure 4a, b should have appeared in the following sentence. It has been now included in the proper position.

**page 5 line 18 – fig implies that hyperhumid does not differ from subhumid;**

Answer: Actually, differences between the hyperhumid and the subhumid locations are significant in 2012, but not in 2013. In the revised version, we rewrote this sentence as follows: "The highest $D_h$ values were found for *Qpyr* at the humid (both years) and subhumid sites (2012), while *Qrob* had a more reduced variation along the gradient (2012 $F_{[2, 117]} = 2.89$, 2013 $F_{[2, 117]} = 0.18$, $P > 0.050$) (Fig. 4b)." (page 5 lines 26-28).

**page 5 line 19 – it is not clear which species are being referred to here;**

Answer: This paragraph was ambiguous. We rewrote it in a clearer form (page 5 lines 32-42).

**page 5 line 26 – clarification is needed here as to what species is being referred to for the budburst range, furthermore, are these comparisons referring to min/max, as the figure implies means of only end of March (90 days) to early May (130);**

Answer: Dates provided in the revised version are those corresponding to min/max values for each species.

**page 5 line 27 – clarify what is being compared here;**

Answer: We rewrote this paragraph as follows: "*Qrob* exhibited an earlier budburst than *Qpyr* ($F_{[1, 236]}$ = 527.83, $P$ < 0.001), occurring from early March to late April for the former, and from mid April to late May for the latter. In both species, budburst occurred earlier at the subhumid site than at humid and hyperhumid locations (Fig. 5b)" (Page 5, lines 35-37).

**page 5 line 29 – this correlation does not appear to be consistent across all sites;**

Answer: The correlation provided was performed after pooling the data from the three study sites. As the referee rightly stated, this correlation was not consistent across sites (Table R1), being thus related to a site effect. Thus, we removed this calculation from the manuscript and the corresponding sentences in Results and Discussion (page 8 lines 27-28 of the former version).

**Table R1** Pearson's correlation coefficients for each species and site. * $P$ < 0.05.

| Spp | Hyperhumid | Humid | Subhumid |
|---|---|---|---|
| *Q. robur* | -0.317* | 0.103 | 0.197 |
| *Q. pyrenaica* | 0.034 | 0.044 | -0.306 |

**page 5 line 34 – what is "ALT" referring to?**

Answer: We apologize for this mistake; we put "ATL" instead of "Hyperhumid site". This error has been corrected in the revised version.

**While the authors acknowledge limitations of this study, in particular the need to include more tissue types for NSC analyses and subsequent comparisons, using only stemwood NSC reserves as a proxy for hypotheses put forth in this research is possibly flawed. A primary concern is that stores of NSC reserves in the root system could have a huge impact on growth, budburst, etc, and cannot be ignored. Such analyses would need to be conducted in order to present this research as acceptable.**

Answer: As the referee pointed out, we are aware of the limitations of our study, which were detailed in the Conclusion section. Despite this, we think that the work presented in our manuscript is novel and supposes an incremental advance in this topic that may be interesting for a broad audience. In this regard, we would like to point out that several anatomical and phenological parameters (vessel size and number, ring width, budburst dates, stem diameter) were assessed simultaneously with carbohydrate reserves for the first time in ring-porous species. In addition, our data set is large, involving two species and three sites along a rainfall gradient (and 240 trees).

Carbon reserves in deciduous trees are mostly stored in the stem, as well as in coarse roots and branches (Barbaroux et al. 2003, *New Phyt*). Probably, additional NSC measurements in roots and branches would have been essential if our objective had been to analyze

changes in the total NSC pool, as it has been done in studies testing the carbon starvation hypothesis (i.e. Anderegg and Anderegg 2013, *Tree Phys;* Galvez et al. 2013*, New Phyt;* Hartmann et al. 2013, *Func Ecol*); or if we had been interested in analyzing carbon fluxes among different tree compartments (Regier et al. 2010, *Tree Phys*). However, although we recognize that NSC measurements in roots would have been valuable in this study, our objective was to disentangle the interaction between xylem growth and short-distance NSC content, rather than focusing on the differences in NSC storage patterns among populations or species.

According to Steppe et al. (2015, *Trends Plant Sci*), a mechanism of xylem growth is dependent on the incorporation of carbon resources, and thus requires information on sugar concentrations in the stem. In this regard, a number of recently published studies analyze the interplay between carbohydrate content and xylem growth by sampling the cambial zone (Deslauriers et al. 2009, *Tree Phys*; Deslauriers et al. 2014, *Ann Bot*), stem sapwood (Galiano et al. 2011, *New Phyt*; El Zein et al. 2011, *Tree Phys*; Oberhuber et al. 2011, *Can J For Res;* Carbone et al. 2013*, New Phyt*), or several above ground compartments (Sala and Hoch et al. 2009, *Plant, Cell & Environ*; Fajardo et al. 2012, *New Phyt*; Saffell et al. 2014, *Tree Phys*). It is interesting to note that NSC translocation through the different plant compartments may be strongly reduced during dormancy if phloem becomes non functional, as suggested by Lacointe et al. (2004, *Plant, Cell & Environ*) for a deciduous walnut. Therefore, the contribution of local stem carbon reserves might probably be considerable in fuelling xylem growth before the sink-to-source transition of leaves (Begum et al. 2010, *Ann Bot*). This is related to the fact that initiation of cambial divisions in roots and stem precedes budburst in deciduous oaks and is consistent with the strong decline in stem NSC concentration frequently reported in spring (Barbaroux and Bréda, 2002, *Tree Phys*; El Zein et al. 2011, *Tree Phys*). For these reasons, we think that relations between stem sapwood NSC content and earlywood growth dynamics are especially relevant.

Hypotheses in page 2 were more accurately defined by specifying that interpretations are only referred to stem sapwood. Thus, we rephrased the sentence page 2, lines 37-38 as follows: "We also hypothesized that the more drought-tolerant *Q. pyrenaica* will have a more reduced xylem growth than *Q. robur*, but larger stem sapwood NSC concentrations". Likewise, in page 2, lines 39-42, we specified more clearly: "(i) stem diameter influences earlywood vessel size, which in turn affects NSC content in the stem; (ii) higher sapwood SS content in winter predisposes trees to advance growth resumption in spring, as well as to produce more earlywood vessels; and (iii) earlywood vessel n and size are key predictors of latewood growth in oaks". In addition, in page 7, lines 13-16, we included a sentence highlighting the need for further research concerning the NSC pool size at the whole tree level (with a special focus on roots) in order to confirm or reject our hypothesis about divergent carbon use strategies in study oaks.

**REFEREE#2**

**Overall, this was an interesting and useful contribution to the ongoing discussion about the roles of NSCs and plant hydraulics on tree phenology, growth, and survival. In this paper the authors studied two congenator oaks of that contrast in their ecological strategies to compare the impacts of winter NSC storage, hydraulic diameter, and budburst on earlywood vessel production (EVP) and the subsequent impacts of EVP, hydraulic diameter, foliar density, growing season length, and NSC on latewood production. Species were evaluated at three sites that form a moisture gradient in northwest Spain.**

**This paper was generally well written and well cited and most of my concerns are moderate and should not change the overall results.**

**Page 2, Lines 22-24: Here you describe one of your study species, but you fail to describe the other. I know Q. robor is more common, but not all your readers will be familiar with its ecology.**

Answer: In the revised version, this paragraph was rewritten as follows: "This is the case of the ring-porous oaks *Quercus robur* L. and *Q. pyrenaica* Willd., which coexist in NW Iberian Peninsula. The former is widespread in Europe, being abundant in areas with mild-oceanic climate. By contrast, *Q. pyrenaica* is dominant in various mountainous ranges of the sub-Mediterranean area, hence exhibiting multiple adaptations to cope with summer drought and winter frost, such as late flushing (Pérez-de-Lis et al., 2016)" (page 2 lines 20-23 of the revised version of the manuscript).

**Page 3, Line 21: No description is given as to HOW the trees were selected. In particular, I have no idea if the authors put out plots of some standard design, picked 'representative' trees, or picked the 40 biggest, healthiest trees they could find. No description is given of the size threshold or other criteria for inclusion (we could in theory be comparing a sapling at one site to a 100cm DBH tree at another). Unfortunately, ample evidence exists to show that trees and locations chosen subjectively to be 'normal' or 'representative' tend to be far better off than random, which unfortunately would cause all of the ANOVA-based comparative analyses to fall into question and require very careful interpretation of the regression-based analyses. I think in any revision the authors need to provide considerable more information about sampling and the editor should pay careful attention this information in assessing the validity of the work. For the remainder of the review I'm going to assume the sampling was done correctly (randomized locations, randomized trees within location).**

Answer: A north-to-south transect was set following a gradient of decreasing humidity. The experiment was carried out at three sites situated along this transect. Both study species were present at the selected locations. At all the sites, *Qrob* and *Qpyr* trees were randomly selected, although highly suppressed and juvenile individuals were disregarded. In the revised version, we included a more detailed description of tree selection in page 3 line 5 and page 3 lines 21-22.

**Page 3, Line 32: How was sapwood area determined?**

Answer: Sapwood can easily be distinguished by colour. Heartwood in oaks is brown-coloured while sapwood has a pale tone (Figure R1). For the sake of clarity, this information was included in the revised version of the manuscript (page 3, lines 35-36).

**Figure R1** Transverse section of a wood core of *Q. pyrenaica* showing the colour boundary between heartwood and sapwood.

[Figure]

**Page 4, Line 37: I'm going to assume growing season length is an individual-level measure and not a site-level measure (as is commonly done), otherwise this effect is confounded with the site random effect.**

Answer: Indeed, growing season length is an individual-level parameter. In the revised manuscript, we cleared up this issue by modifying the paragraph regarding the methodology for phenological monitoring (page 3 lines 24-26).

**Page 5, Lines 1-2: Here you're talking about averaging over a set of models, but in the paragraph above you only describe a single model. Where does this other set of**

**models come from? Why do you need another set of models? Why is the sum of Akaike weights an appropriate measure of the relative importance of a variable? This quantity is quite challenging to interpret, especially in a GLMM, and fairly unintuitive. I'm all for sophisticated analyses when needed, but why not stick to a simpler analysis (e.g. the proportion of the variance [R2] explained by each covariate),which in my mind would be much easier to interpret and a more direct measure of importance. As I tend to look at the figures before I read a paper, I'll also note that the meaning of 'relative importance' (essentially a weighted number of times that a variable was included in the model) is not clear in the figure.**

Answer: What we meant with "set of models" is that we calculated the AIC of the models containing all the possible fixed-effect combination. In the former version we used an information-theoretic approach to identify the most influent fixed effects of the model. According to this procedure, models were compared using their AIC scores (the lower the AIC, the better the model fit). Hence, models were ranked and averaged in order to assess the relative weight of each variable (we averaged 95% of all the fitted models according to their AIC scores). As we mentioned in the former version, this method was detailed in Burnham and Anderson (2002), and has been used in a recent paper analyzing possible limitations of carbon supply on secondary growth published in Biogeosciences (Guillemot et al. 2015). In order to provide more confident results, we included the estimates, variance partitioning, the variance explained by each fixed factor (as a percentage of the total variance explained by the fixed effects), proportion change in variance, and $R^2$ for mixed-effects models (marginal and conditional). This information is presented in Table 2 of the revised version.

**Page 5, Line 8: You should report the degrees of freedom in the F test (and all other tests). If this is going to be the same for all subsequent analyses state that here at the first usage, otherwise make the df explicit for each analysis.**

Answer: We agree to the referee. *df* values were included in brackets next to $F$, $\chi^2$, and *r*.

**Page 5, Line 34: Be consistent with notation. In all other places you refer to sites by their moisture status, and here you've reverted to a site code, and I'm not sure which site you're referring to.**

Answer: We apologize for this mistake again; we put "ATL" instead of "Hyperhumid site". This error was corrected in the revised version.

**Page 5, Line 35: Were trees with powdery mildew included or excluded? Why wasn't this included as a covariate? Why is there not more in the discussion about how this could be affecting results?**

Answer: Unfortunately, powdery mildew infestation has not been quantified. Yet, it is relevant to take into account that all the trees were more or less affected by the pest (it would not be considered as a covariate, but as a part of the site effect). This is a very frequent disease at oak forests in the study region, but their effects during the humid spring of 2013 were higher than usual at the hyperhumid location. Thus, we decided to provide this information, which probably helped us to interpret the lower growth noted at this latter site. The possible effect of powdery mildew infestation is discussed in page 6 line 40 of the revised version. Since we did not perform any measurement, we marginally commented this issue.

**Page 5, Line 37 to Page 6, Line 3: In the Results (here) and Discussion (below), I'm concerned that the authors are over-interpreting the biological significance of results that are statistically significant but have low R2. Looking at Figure 5, about all I'm comfortable concluding is that SS and tree size have a negative impact on budburst in both species, and that SS had a positive impact on EVP in Q. pyrenaica. Effects in the**

**R2 of 3-6% range (Starch, Q robor EVP) don't seem worth discussing, and those in the 10-16% range (Dh, SS) should be acknowledged as weak.**

Answer: Thanks for the comment. In the revised version, we included a more careful interpretation of the relationships accounting for a low variance (in both SEM and GLMM) in the observed parameters with several changes in the Results and Discussion sections (particularly in Subsections 3.2, 4.2, and 4.3).

**Page 6, line 31: Tree density effects are speculative**

Answer: We acknowledge that direct measurements on tree competition were not carried out. But this idea was based on differences in stand tree density (reported in Pag 3 line 13) and basal area (according to stem diameter measurements in Figure 5a) among locations. In this regard, recent work modelled a strong effect of competition in *Q. pyrenaica* secondary growth (Fernández-de-Uña et al., 2016). For the sake of clarity, we rearranged the paragraph concerning differences among sites in subsection 4.1. In addition, the sentence in page 6 lines 35-36 of the revised version was rewritten as follows: "
[revised manuscript text omitted]